# Does Dietary Lipid Level Affect the Quality of Triploid Rainbow Trout and How Should It Be Assessed?

**DOI:** 10.3390/foods12010015

**Published:** 2022-12-21

**Authors:** Yuqiong Meng, Xiaohong Liu, Lingling Guan, Shoumin Bao, Linying Zhuo, Haining Tian, Changzhong Li, Rui Ma

**Affiliations:** 1State Key Laboratory of Plateau Ecology and Agriculture, Qinghai University, Xining 810016, China; 2College of Ecological Environmental Engineering, Qinghai University, Xining 810016, China

**Keywords:** appearance quality, texture, odor, taste, nutrition, multivariate analyses

## Abstract

Organoleptic properties and nutritional value are the most important characteristics of fish fillet quality, which can be determined by a series of quality evaluation indexes and closely related to fish nutrition. Systematic organoleptic and nutritional quality evaluation indexes consisting of 139 indexes for physical properties and chemical compositions of triploid rainbow trout were established. Besides, effects of dietary lipid levels (6.6%, 14.8%, 22.8% and 29.4%) on the quality of triploid rainbow trout were analyzed in the study. The main results showed that, for fillet appearance quality, fish fed diets with lipid levels above 22.8% had higher fillet thickness and redness but lower gutted yield and fillet yield (*p* < 0.05). For fillet texture, fish fed the diet with a 6.6% lipid level had the highest fillet hardness (5.59 N) and lowest adhesiveness (1.98 mJ) (*p* < 0.05), which could be related to lipid, glycogen, water soluble protein and collagen contents of the fish fillet. For fillet odor, the odor intensity of “green, fatty and fishy” significantly increased with the increase of the dietary lipid level (from 1400 to 2773 ng/g muscle; *p* < 0.05), which was related to the degradation of n-6 and n-9 fatty acids. For fillet taste, a high lipid diet (≥22.8%) could increase the umami taste compounds contents (from 114 to 261 mg/100 g muscle) but decrease the bitterness and sourness taste compounds contents (from 127 to 106 mg/100 g muscle and from 1468 to 1075 mg/100 g muscle, respectively) (*p* < 0.05). For nutritional value, a high lipid diet could increase the lipid nutrition level (such as the content of long chain polyunsaturated fatty acids increased from 3.47 to 4.41 g/kg muscle) but decease tryptophan and selenium content (from 2.48 to 1.60 g/kg muscle and from 0.17 to 0.11 g/kg muscle, respectively). In total, a high lipid diet could improve the quality of triploid rainbow trout. The minimum dietary lipid level for triploid rainbow trout should be 22.8% to keep the better organoleptic and nutritional quality.

## 1. Introduction

Rainbow trout (*Oncorhynchus mykiss*) is a globally farmed species. To avoid the degradation of fish quality associated with sexual maturation, triploid rainbow trout has been the main cold-water fish species cultured in China [1]. With the increase in production, fillet quality has become an important factor to determine consumer preference and sustainable development of aquaculture for the fish.

Consumer preference of fish quality is determined mainly by the organoleptic properties and nutritional value, and both can be reflected by fish physical properties and chemical compositions of fish [2]. Fish products are recommended for human consumption for the nutritional value and health attribute, mainly including protein, essential amino acid (EAA), n-3 long-chain polyunsaturated fatty acids (n-3 LCPUFA) [3,4,5]. The organoleptic properties of meat are associated with human sensory perception (visual, smell, taste and mouthfeel sense), consisting of appearance, color, texture and flavor. Sensory attributes are traditionally evaluated by the human sensory panel, which takes a lot of time and effort for training [6]. Compared with the sensory test, the analysis of related physical properties and chemical compositions has the advantages of being reliable, simple, cost-effective and applicable to a wide range of food products with minimum false results [7]. Therefore, it is urgent to establish an evaluation system of fish quality on the basis of analytical techniques for triploid rainbow trout. This will help to find the relationship between the abstract quality concept and the specific concentration or presence of certain species. Finally, it will lay the foundation for fish quality control.

Fish nutrition and feeds is a main affecting factor of fish quality, together with fish intrinsic characteristics, environmental factors, procedures of harvesting and post-harvesting [2]. Acting as an important macronutrient in fish feed, lipids not only provide essential fatty acids for fish health and growth but also can be used as a nonprotein energy source which can effectively reduce nitrogen and phosphorus loads produced by high-protein diets [8,9]. At present, high lipid diets are used in salmonids aquaculture. It has been shown in previous studies that flesh quality, including fillet color, texture, flavor and nutritional compositions, can be altered by high dietary lipid levels [8,10,11]. Our previous study has found that triploid rainbow trout could tolerate dietary lipid levels above 22.8% with no effects on fish growth and health [9], however, does high lipid diets improve or reduce the organoleptic properties and nutritional value of triploid rainbow trout is a question remains to be answered.

Thus, the present study used multiple indices related to fillet physical properties and chemical compositions to evaluate fish organoleptic properties (including appearance quality, texture, odor and taste) and fish nutritional value (including protein and amino acids, lipid and fatty acids and mineral nutrients). The analytic techniques were used to study systematically the quality of triploid rainbow trout affected by dietary lipid levels.

## 2. Materials and Methods

### 2.1. Animals, Experimental Diets and Feeding Trial

The present study was performed in strict accordance with the Standard Operation Procedures of the Guide for the Use of Experimental Animals of Qinghai University. The research protocol was reviewed and approved by the Ethical Committee of Qinghai University. Female triploid rainbow trout were obtained from local fisheries (Qinghai, China).

Four iso-proteinic experimental diets (46% dry matter) with graded lipid levels of 6.6, 14.8, 22.8 and 29.4% were specifically formulated and presented in Appendix A. For the feeding trial, each of the four experimental diets was randomly fed to four replicate net cages (*n* = 4) for 80 days. The feeding strategy, aquaculture management and water quality conditions were reported in our previous study [9]. After the feeding trail, the fish average weight of the 6.6, 14.8, 22.8 and 29.4% group reached 468, 610, 655 and 682 g, respectively, from the same initial average weight of 233 g [9].

### 2.2. Sample Collection

After 3 days of fasting at the end of the feeding trial, three experimental fish were randomly selected in each net cage and euthanized in excess anesthetic based on the method described in a previous study [12]. There were 12 fish in each dietary lipid level group (*N* = 12). All the experimental fish were weighed and measured individually, and bled by gill cut in an ice slurry for at least 10 min. Subsequently, the viscera were removed and the weighed carcass gutted. After that, both fish fillets were filleted and measured.

The left side fillet of each fish was packaged and stored in an ice box for 48 h. After transportation to the laboratory of Qinghai University, texture, water holding capacity (WHC), pH and color of the fillets were tested immediately. Then, the fillets were divided into several segments and stored at −80 °C to analyze the other quality indicators. Specific segments or locations on each fillet where the analyses were performed are found in Figure 1.

### 2.3. Fillet Appearance Quality Analysis

Based on the weight and length of both fillets, gutted yield (GY), fillet yield (FY) and relative fillet length (RFL) were calculated as:(1)GY (%)=100 × ((carcass gutted weight (g)/body weight (g))
(2)FY (%)=100 × (fillets weight (g)/body weight (g))
(3)RFL=fillet length (cm)/body length (cm)

Fillet thickness was assayed by a texture analyzer (TMS-PRO, FTC, Sterling, VA, USA) on the basis of the texture profile analyses (TPA).

Fillet color per each fillet was determined by *L** (lightness), *a** (redness) and *b** (yellowness) directly using a colorimeter (CR-400, Minolta, Tokyo, Japan). The chroma (*C*_ab_*) and hue ($H_{\mathrm{ab}}^{0}$) were calculated based on the *a** and *b** values as:(4)$$C_{\mathrm{ab}}^{*}{=(a}^{*}{}^{2}{+b}^{*}{}^{2}{)}^{1/2}$$
(5)$$H_{\mathrm{ab}}^{0}{=\arctan\;(b}^{*}\;/{\;a}^{*})$$

### 2.4. Texture, WHC, pH, Moisture, Ash, Protein, Water and Salt Soluble Protein, Collagen and Glycogen Analysis

Fillet texture was evaluated by TPA using a texture analyzer (TMS-PRO, FTC, Sterling, VA, USA) equipped with a 25 kg load cell and an 8 mm cylindrical probe, as described in a previous study [13].

The WHC was defined by measuring the expressible moisture using a texture analyzer (TMS-PRO, FTC, Sterling, VA, USA) according to Schubring et al. [14] with some modifications. Muscle sample was flatted between paired filter papers and then squeezed by texture analyzer equipped with a 25 kg load cell and a 38 mm cylindrical probe. The detection rate was set to 30 mm/min. The sample was pressed with a force load of 1 kg and held at that point for 1 min. WHC was calculated as:(6)WHC (%)=100 × ((initial weight (g) − final weight (g))/initial weight (g)

The pH was measured using a solid electrode (Inlab^®^ Solids Pro-ISM, Mettler Toledo, Zurich, Swiss) connected to a pH-meter (S220, Mettler Toledo, Zurich, Swiss), which was directly inserted into the fillet.

The moisture, ash, protein, water and salt soluble protein (WSP and SSP), as well as collagen (measuring hydroxyproline content, HYP) contents in the fillet samples, were assayed using the methods described in our previous study [13]. Glycogen and pyridinoline crosslink (PYD) content was measured by a commercial assay kit using the anthrone-sulfuric acid colorimetric method (Nanjing Jiancheng Bioengineering Institute, Nanjing, China) and ELISA assay method (Shanghai Enzyme-linked Biotechnology Co., Ltd., Shanghai, China), respectively.

### 2.5. Lipid and Fatty Acids Analysis

The lipid content was determined by the chloroform-methanol extraction method [15] and the extracted lipid was subsequently used for fatty acid analysis. The methyl esterification and analysis of the fatty acids were based on the method described by Ma et al. [13] using a gas chromatography and mass spectrogram (GC-MS; QP2020, Shimadzu, Kyoto, Japan) with Rxi-5 sil MS (30 × 0.25 mm, 0.25 μm). Fatty acids were identified by the external standards (37 Component FAME Mix, Supelco, Bellefonte, PA, USA) and quantified by the internal standard concentration (methyl heptadecanoate, Sigma, Shanghai, China). The atherogenic index (AI) and thrombogenic index (TI) used as evaluating fat quality indices [16] were calculated as follows:(7)$$\mathrm{AI}=\frac{\left(C12:0+4\;\times \;C14:0+C16:0\right)}{\left(\Sigma \mathrm{MUFAs}+\Sigma \mathrm{PUFAs}\right)}$$
(8)$$\mathrm{TI}=\frac{\left(C14:0+C16:0+C18:0\right)}{(0.5\;\times \;\Sigma \mathrm{MUFAs}+0.5\;\times \;\Sigma n\text{-}6\;\mathrm{PUFAs}+3\;\times \;\Sigma n\text{-}3\;\mathrm{PUFAs}+\Sigma n\text{-}3\;\mathrm{PUFAs}/\Sigma n\text{-}6\;\mathrm{PUFAs})}$$
where MUFA and PUFA represent monounsaturated fatty acid and polyunsaturated fatty acids, respectively.

### 2.6. Volatile Compounds Analysis

The volatile compounds in fillets were assayed by the method described in our previous study [17] using the gas chromatography-mass spectrometry (GC-MS, QP2020, Shimadzu, Japan) coupled with automated solid-phase microextraction (SPME) system (AOC-6000, CTC, Zwingen, Switzerland). Volatile compounds were identified by chemical standards (Sigma, Shanghai, China) and linear retention indices (calculated using a series of n-alkanes). The concentrations were quantified by the ratio of the peak area to the internal standard (2, 4, 6-trimethyl-pyridine, Sigma, Shanghai, China). Odor activity value (OAV) was used to evaluate the contribution of each volatile compound. When the OAV was equal to or greater than 1, it was considered that the substance was an odor-active compound contributing to the overall aroma [18]. The OAV was calculated as:(9)$${\mathrm{OAV}=C}_{i}/{\mathrm{OT}}_{i}$$
where C_i_ is the concentration of the volatile compound, OT_i_ is the odor threshold of the compound reported in the literature [17].

### 2.7. Amino Acids Analysis

Free amino acid was extracted and measured by the method described by Chen and Zhang [19] using a high performance liquid chromatograph (HPLC; 1260 Infinity II, Agilent, Santa Clara, CA, USA).

Three hydrolysis methods were used to determine total amino acids contents. The determination of methionine and cystine was based on the methods of oxidation with performic acid and hydrolysis [20]. The determination of tryptophan was based on the method of alkaline hydrolysis [21]. The determination of other amino acids was based on the method of acid hydrolysis [3]. Except tryptophan, the other amino acids were analyzed by an amino acid analyzer (S433D, SYKAM, Munich, Germany). The tryptophan was analyzed by HPLC (1260 Infinity II, Agilent, Santa Clara, CA, USA).

On the basis of the adult essential amino acid requirements, with respect to the WHO/FAO/UNU [22]. The requirement of each essential amino is histidine 15, isoleucine 30, leucine 59, lysine 45, methionine + cysteine 22, phenylalanine + tyrosine 38, threonine 23, valine 39, tryptophan 6 mg/g protein, respectively), essential amino acid score (EAAS) was calculated using the following formula:(10)$$\mathrm{EAAS}=100\;\times \;\frac{\mathrm{essential}\;\mathrm{amino}\;\mathrm{acid}\;\mathrm{content}\;\mathrm{of}\;\mathrm{the}\;\mathrm{sample}}{\mathrm{essential}\;\mathrm{amino}\;\mathrm{acid}\;\mathrm{requirement}\;\mathrm{for}\;\mathrm{adult}\;}$$
where the amino acid content was expressed as mg individual amino acid per g total protein.

### 2.8. Nucleotides Analysis

The nucleotides were extracted and analyzed by the method described in our previous study [13] using HPLC (1260 Infinity II, Agilent, Santa Clara, CA, USA) coupled with a CAPCELL PAK C18AQ S5 column (4.6 mm × 250 mm, 5 μm, OSAKA SODA, Osaka, Japan). Guanosine-5′-monophosphate (GMP), adenosine-5′-monophosphate (AMP), inosine-5′-monophosphate (IMP) were identified and quantified by the external standard curve (Sigma, Shanghai, China).

Based on the description of Liu et al. [23], the equivalent umami concentration (EUC) was calculated by free amino acids (free aspartic acid and glutamic acid, Free asp and Free glu) and 5′-nucleotides (IMP, GMP and AMP) using the following equation:(11)$$\mathrm{EUC}\;(\mathrm{gMSG}/100\;g)=\Sigma a_{i}\;+\;1218\;(\Sigma a_{i}b_{i})\;(\Sigma a_{j}b_{j})$$
where *a_i_* is the concentration (g/100 g) of each umami amino acid in terms of Free asp or Free glu; *b_i_* is the relative umami concentration (RUC) for each umami amino acid versus monosodium glutamate (MSG; Free asp, 0.077 and Free glu, 1); *a_j_* is the concentration (g/100 g) of each umami 5′-nucleotide (IMP, GMP and AMP); *b_j_* is the RUC for each umami 5′-nucleotide versus IMP (IMP, 1; GMP, 2.3 and AMP, 0.18); 1218 is a synergistic constant.

### 2.9. Organic Acids Analysis

The organic acids were extracted and analyzed using the procedure described by Liu et al. [23] with some modification. In brief, one gram muscle sample was homogenized in 5 mL purified water and centrifuged at 10,000× *g* for 15 min, and then filtered through a 0.22 μm membrane prior to HPLC analysis (LC-20AT, Shimadzu, Kyoto, Japan) equipped with a CAPCELL PAK C18 MG column (4.6 mm × 250 mm, 5 μm, OSAKA SODA, Osaka, Japan). The HPLC conditions were as follows: column temperature was 35 °C and flow rate was 1.0 mL/min. Mobile phase was the mixture of methanol (A) and 0.05% phosphoric acid (B) using the following gradient: 0~3 min, 1% A~5% A; 3~8 min, 5% A~23% A; 8~12 min, 23% A~50% A; 12~15 min, 50% A~100% A. Detector wave length was set at UV 214 nm. Each organic acid was identified and quantified by the retention time and standard curve (21 species of organic acid standards kit, Supelco, Bellefonte, PA, USA).

### 2.10. Inorganic Ions Analysis

Based on the procedure described by Erkan and Özden [24], the concentrations of calcium (Ca), magnesium (Mg), potassium (K), sodium (Na), zinc (Zn), copper (Cu), iron (Fe), manganese (Mn) and selenium (Se) in the fish fillet were determined using an inductively coupled plasma mass spectrometer (ICP-MS; ICAP RQ, Thermo, Waltham, MA, USA) after being digested with 8 mL of HNO_3_ (65%) and 2 mL of H_2_O_2_ (30%) in a microwave oven decomposition system (MARS 6, CEM, Matthews, NC, USA). The nine standard mixed solution for ICP analysis (GRINM, Beijing, China) was diluted in different concentrations and used to calibrate the ICP-MS. The germanium and scandium standard mixed solution (30 μg/L; GRINM, Beijing, China) was used as internal standard.

Based on the method described by Liu et al. [23] with some modification, the phosphate and chloride concentrations were measured by using an ion chromatograph system (ICS-1100, Dionex, Sunnyvale, CA, USA) equipped with an anion pre-column (IonPac AG23, 4.0 mm × 50 mm), an anion exchange analytical column (IonPac AS23, 4.0 mm × 250 mm), a suppressor (Dionex ADRS 600, 4 mm) and a conductivity detector. The sample (1.0 g) was homogenized in 50 mL deionized water, and then extracted by ultrasound for 30 min. After a water bath at 75 °C for 5 min, the mixture was made up with deionized water to a 50 mL flask. Some solution after filtering with filter paper was centrifugated at 10,000× *g* for 15 min and filtered through a 0.45 μm membrane and Dionex OnGuard^TM^ II RP column prior to injection into the ion chromatography system. The ion chromatography conditions were as follows: the flow rate was 1.0 mL/min at room temperature. The isocratic elution was the mixture of 4.5 mmol/L Na_2_CO_3_ and 0.8 mmol/L NaHCO_3_.

### 2.11. Taste Activity Value and Nutritional Contribution

Taste activity value (TAV) was used to evaluate the contribution of each taste compound (including free amino acids, 5′-nucleotides, organic acids or inorganic ions) in the fish fillet. When the TAV was equal to or greater than 1, it was considered that the substance was a taste-active compound contributing to the food taste. The TAV was calculated as:(12)$${\mathrm{TAV}=C}_{i}/{\mathrm{TT}}_{i}$$
where C_i_ is the concentration of the taste-related compound, and TT_i_ is the taste threshold of the compound reported in the literature [23].

Nutritional contribution (NC) value of fish fillet was calculated as described by Ramalho Ribeiro et al. [4] using the following equation:(13)NC (%)=100 × (C × M)/(DRI or DAI)
where C: concentration of nutrient in g/kg, mg/kg or μg/kg. M: meal fillet portion consumed is 0.16 kg. The daily recommended intake (DRI) established as the recommended intake for protein, linoleic acid, α-linolenic acid to adolescents and adults (mean value of man and woman) is 47.8 g/d, 13.1 g/d and 1.3 g/d, respectively [25]. DRI established as the recommended intake for Mg, Mn, Fe, Zn, Cu and Se to adolescents and adults (mean value of man and woman) is 348 mg/d, 2.0 mg/d, 10.5 mg/d, 9.3 mg/d, 865 μg/d and 52.5 μg/d, respectively [26]. DRI established as the recommended dietary allowances and adequate intake for Ca to adolescents and adults (mean value of man and woman) is 1150 mg/d [27]. DRI established as the recommended dietary allowances and adequate intake for Na and K to adolescents and adults (mean value of man and woman) is 1450 mg/d and 2842 mg/d [28]. The daily adequate intake (DAI) for EPA + DHA is 500 mg/d for primary prevention of cardiovascular disease in adults recommended by the International Society for the Study of Fatty Acids and Lipids (ISSFAL) [4].

### 2.12. Statistical Analysis

One-way ANOVA was conducted using SPSS 19.0 software. When significant differences (*p* < 0.05) were observed, Turkey’s multiple range test was used to compare differences among four dietary lipid levels groups. A total of 139 fillet quality indicators were used for the principal component analysis (PCA) to assay the relationship among indicators and classify the different dietary lipid level groups in company with cluster analysis. The heatmap and enrichment analysis were performed by MetaboAnalyst 5.0 online software (https://www.metaboanalyst.ca/ (McGill University, Montreal, QB, Canada) (accessed on 16 September 2022)) based on the quality indicators with significant difference and the quality indicators of compounds or metabolites, respectively.

## 3. Results

### 3.1. Fillet Appearance Quality

The biometrical parameters and color of the fillet are shown in Table 1. GY and FY decreased significantly when the dietary lipid level increased up to 22.8% (*p* < 0.05) and then plateaued (*p* > 0.05), while FT showed an opposite tendency (*p* < 0.05). Fish fed the diet containing lipid of 6.6% had the highest RFL (*p* < 0.05). For fillet color, *L**, *b** and *C*_ab_* values showed no significant difference (*p* > 0.05). *a** value and $H_{\mathrm{ab}}^{0}$ value increased and decreased, respectively, when the dietary lipid level increased up to 22.8% (*p* < 0.05) and then remained consistent (*p* > 0.05).

### 3.2. Fillet Texture

Data on physical properties and biochemical compositions are shown in Table 2. No difference was found in fracturability, cohesiveness, springiness, chewiness and WHC among dietary lipid levels groups (*p* > 0.05). Fish fed the diet with the lowest dietary lipid level of 6.6% had the highest hardness value (*p* < 0.05). Fillet adhesiveness value and pH increased with an increasing level of dietary lipid up to 29.4% and 22.8%, respectively.

As the dietary lipid level increased, muscle lipid content significantly increased but moisture and glycogen contents decreased (*p* < 0.05). No difference was found in ash content (*p* > 0.05). For protein composition, WSP content decreased when the dietary lipid level increased, while SSP content significantly increased when the level of dietary lipid increased up to 22.8% and decreased thereafter (*p* < 0.05). Fish fed the diet with the 6.6% lipid level had higher a-i HYP and total HYP contents than fish fed the diets with 14.8%, 22.8% and 29.4% (*p* < 0.05), while a-s HYP and PYD contents were not affected by dietary lipid levels (*p* > 0.05).

### 3.3. Fillet Odor

Based on the description of OAV, 15 odor-active compounds were identified in raw fillet of triploid rainbow trout by using SPME-GC-MS (Table 3). The concentrations of 2-octen-1-ol, hexanal, 2-hexenal, (*E*)-2-heptenal, (*Z*)-4-heptenal and sum of volatiles derived by n-3 fatty acids (∑n-3 derived) increased with the dietary lipid level up to 14.8% (*p* < 0.05) and plateaued thereafter (*p* > 0.05). Concentrations of octanal and sum of volatiles derived from n-6 fatty acids (∑n-6 derived) increased with the level of dietary lipid up to 22.8% (*p* < 0.05) and then plateaued (*p* > 0.05). The concentrations of 1-heptanol, 1-octen-3-ol, pentanal, heptanal, (*E, E*)-2,4-heptadienal, (*E*)-2-octenal, nonanal, (*E*)-2-nonenal, (*E, Z*)-2,6-nonadienal, sum of volatiles derived by n-9 fatty acids (∑n-9 derived) and total concentration of 15 odor-active compounds (TOAV) increased with the dietary lipid level up to 29.4% (*p* < 0.05).

### 3.4. Fillet Taste

Data on the taste-related compounds with umami, sweet, bitter, sour and salty are shown in Table 3. For umami taste compounds, AMP, IMP and GMP contents increased with an increasing level of dietary lipid up to 22.8% (*p* < 0.05) and then decreased (*p* < 0.05). Fish fed the diet with the 6.6% lipid level had the lowest content of Free asp, which was significantly lower than fish fed the diet with the 14.8% lipid level (*p* < 0.05), however, Free glu content was similar in all groups (*p* > 0.05). In total, the sum content of umami taste compounds (SUTC) and EUC increased as the dietary lipid level increased to 22.8% (*p* < 0.05) and then plateaued (*p* > 0.05). For sweet taste compounds, 6 free amino acid species related to sweet taste, consisting of threonine (Free thr), serine (Free ser), glycine (Free gly), alanine (Free ala), proline (Free pro) and lysine (Free lys), showed different trends with an increasing level of dietary lipid, resulting in the sum content of sweet taste compounds (SWTC) not being affected by the dietary lipid level (*p* > 0.05). For bitter taste compounds, the contents of seven free amino acid species, consisting of arginine (Free arg), valine (Free val), methionine (Free met), leucine (Free leu), isoleucine (Free iso), phenylalanine (Free phe), tyrosine (Free tyr) except histidine (Free his) related to bitter taste, were generally lower in fish fed the diets with 22.8% and 29.4% lipid levels than fish fed the diets with 6.6% and 14.8% lipid levels (*p* < 0.05). In total, fish fed the diet with the 22.8% lipid level had the lowest sum content of bitter taste compounds (SBTC), no difference was observed between 22.8% and 29.4% groups (*p* > 0.05). For sour taste compounds, only lactic acid, succinic acid, oxalic acid and maleic acid were identified in the fillet of triploid rainbow trout. No difference in maleic acid content was found among all groups (*p* > 0.05). Fish fed the diet with the lowest lipid level (6.6%) had the highest lactic acid, succinic acid and oxalic acid content as well as the sum content of sour taste compounds (SOTC). For salty taste compounds, the concentrations of Na^+^, K^+^, Ca^2+^, Mg^2+^, Cl^−^, PO_4_^3−^, as well as the sum content of salty taste compounds (SATC), were not affected by dietary lipid levels (*p* > 0.05).

### 3.5. Fillet Nutrition Value

The effects of dietary lipid levels on protein, amino acids, fatty acids and minerals contents of fillet are shown in Table 4. Fillet protein and tryptophan (Trp) content significantly decreased with the increasing level of dietary lipid up to 29.4% and 22.8%, respectively, while no change was observed in other amino acids contents (*p* > 0.05). Fish fed the diets with 22.8% and 29.4% lipid levels had the lowest essential amino acid (EAA) and total essential amino acid (TAA) contents (*p* < 0.05). However, no-essential amino acid (NEAA) content and the ratio of EAA/TAA were not influenced by dietary lipid level (*p* > 0.05). As dietary lipid levels increased, fillet C18:3n-3 and EPA + DNA contents, as well as sum content of polyunsaturated fatty acids (ΣPUFA), sum content of long chain polyunsaturated fatty acids (ΣLC-PUFA), sum content of n-3 polyunsaturated fatty acids (Σn-3) and sum content of n-6 polyunsaturated fatty acids (Σn-6), were significantly increased (*p* < 0.05). Fish fed the diet with the 6.6% lipid level had the lowest C18:2n-6 content and sum content of saturated fatty acids (ΣSFA). However, C18:1n-9, sum content of monounsaturated fatty acids (ΣMUFA) and sum content of n-9 monounsaturated fatty acids (Σn-9) were not affected by dietary lipid level (*p* > 0.05). AI value (ranged 0.27–0.33) but not TI value (ranged 0.36–0.42) was affected by dietary lipid level, both were lower in 22.8% and 29.4% groups (*p* < 0.05). For minerals, Na, K, Ca, Mg, Mn, Fe, Cu and Zn contents were not altered by the dietary lipid level (*p* > 0.05). The lowest Se content of fillet was shown in the highest dietary lipid level group (29.4%), which was significantly lower than that in the 22.8% group (*p* < 0.05).

### 3.6. Multivariate Analyses and Enrichment Analysis

The PCA correlation loading plot and score plot are shown in Figure 2. The 49.1% of the variation in the PCA model was explained by PC1 (38.8%) and PC2 (10.3%). In the PCA correlation loading plot (Figure 2a), each arrow pointed to one quality indicator. Indicators located closely to each other had a positive correlation, while those with loadings of opposite signs were negatively correlated. In the PCA score plot (Figure 2b), all samples were separated into three clusters based on cluster analysis, namely 6.6%, 14.8% and above 22.8% dietary lipid level groups. The results of the quality indicators with significant difference and the overall trend are shown in Figure 2c by heatmap analysis. Based on the results of PCA and cluster analysis, and enrichment analysis based on the KEGG database was used to compare the differences in the metabolic pathway between the group with a dietary lipid level higher than or equal to 22.8% (22.8% and 29.4% groups) and the group with a dietary lipid level lower than 22.8% (6.6% and 14.8% groups). Eleven metabolic pathways with significant difference were found in this study (*p* < 0.05; Figure 2d), including three fatty acids metabolic pathways (α-linolenic acid metabolism, linoleic acid metabolism and biosynthesis of unsaturated fatty acids), two glycometabolism pathways (glycolysis/gluconeogesis and pyruvate metabolism), two amino acids metabolic pathways (tryptophan metabolism, alanine, aspartate and glutamate metabolism) and four other metabolic pathways (purine metabolism, citrate cycle, propanoate metabolism and butanoate metabolism).

## 4. Discussion

### 4.1. Fillet Appearance Quality

Fillet biometrical parameters were important indicators in fish product processing. The present study chose gutted yield, fillet yield, relative fillet length and fillet thickness to evaluate the effects of dietary lipid levels. In the present study, fish fed the diet with a low dietary lipid level obtained a higher gutted yield and fillet yield, which could be related to the lipid deposition pattern. Visceral cavity was the main lipid depots for salmonids; the high dietary lipid level could increase viscerosomatic index [8]. It was also the reason for the diminishing gutted yield of the high lipid level groups in the present study. Based on PCA and heatmap analysis, fillet yield positively correlated with fillet hardness, which could be explained by which the muscle with a high hardness characteristic led to filleting easily. Fish fed diets with 22.8% and more lipid levels showed the short and thick fillet appearance in the present study, which was related to the high fish condition factor (the ratio of body weight to the third power of body length). The positive correlation between condition factor and dietary lipid level in rainbow trout was found in previous studies [29].

The red flesh color is of great importance in consumer acceptance of salmonids quality. The present study showed that diets with a lipid level of 22.8% and 29.4% could increase fillet redness of triploid rainbow trout. Similar results were also found in other studies [8,10,30]. Hence, a high lipid diet (≥22.8%) could promote the deposition of fat-soluble pigment (astaxanthin) and then improve the appearance quality of fish.

### 4.2. Fillet Texture

Texture is an important attribute of flesh quality in fish. According to TPA analysis, the present study found that only hardness and adhesiveness were affected by the dietary lipid level. Decreasing flesh hardness by a high lipid diet was found in another study on rainbow trout [8]. The highest fillet hardness value in the lowest dietary lipid level group was also found in this present study. Previous studies showed that hardness or firmness of the fish fillet was influenced by the flesh lipid content [13]. In the present study, the fillet lipid content increased as the dietary lipid level increased up to 29.4%. However, there was no change in hardness as the dietary lipid level increased from 14.8% further up to 29.4%. The flesh lipid content did not show a clear correlation with the fillet hardness parameter when fish fed the diets with a lipid level above 14.8% in rainbow trout. Similar results were also found in some research of salmonids [10,30]. Based on PCA and heatmap analysis, fillet hardness positively correlated with fillet protein content, especially with collagen content and maturity. The contents of a-i HYP and total HYP represented matured collagen and total collagen, respectively. It might be a reason for the highest fillet hardness value in the lowest dietary lipid level group. Except hardness, fillet adhesiveness increased as the dietary lipid level increased in triploid rainbow trout. Based on PCA and heatmap analysis, the change of adhesiveness was related to the increase of the flesh lipid content, and the decreasing of moisture and water soluble protein contents. Although the adhesiveness was affected by the dietary lipid level, the water holding capacity was not different among all the groups. In total, the muscle with too high hardness and too low lipid content in the 6.6% dietary lipid level group, generally, showed shriveled or inelastic characteristics.

The pH was closely related to the post-mortem evolution of the flesh and played an important role in the texture of the fish fillet [31]. In this study, all fish fillets were sampled following the same procedure in order to keep the same phase of the post-mortem degradation. Why was the lower flesh pH found in the 6.6% lipid level group? Previous study showed that reduction in pH was accompanied by enhanced glycogen degradation and lactate accumulation [32]. In the study, higher glycogen and lactic acid contents were found in the 6.6% lipid level group. It indicated that fish fed the diet with the 6.6% lipid level had higher flesh glycogen content and resulted in higher lactic acid content by anaerobic glycolysis. It was also supported by the result of enrichment analysis that glycolysis and pyruvate metabolism were different between the <22.8% dietary lipid levels group and ≥22.8% dietary lipid levels group.

### 4.3. Fillet Odor

As an important quality indicator, fish odor also determined consumer acceptance and preference. Volatile compounds determine the aroma/odor of food products [33]. To evaluate the overall odor, the OAV was applied in the fish assay based on the content of volatile aromas compounds [18]. In our previous study, 21 odor-active compounds (OAV equal to or greater than 1) were found in raw flesh of triploid rainbow trout [13], however, only 15 species (12 species volatile aldehydes and three species volatile alcohol) were found in this study. The difference might be related to feed composition. The total concentration of 15 odor-active compounds increased, which suggested that as the dietary lipid level increased, the fillet odor intensity increased. The top five compounds in terms of OAV were (*E*)-2-nonenal, 1-octen-3-ol, nonanal, octenal and hexanal, which generally smelled “green, fatty and fishy”. Volatile aldehydes and alcohols were mainly produced by enzymatic decomposition (lipoxygenases) of polyunsaturated fatty acids [34]. Based on the data of Σn-3 derived, Σn-6 derived and Σn-9 derived, the odor-active compounds were mainly produced by n-6 fatty acids, then n-9 fatty acids and finally n-3 fatty acids. Turchini et al. (2007) [35] showed that Σn-3 derived volatiles and Σn-6 derived volatiles were positively correlated with the content of n-3 PUFA and n-6 PUFA, respectively. Similar correlations were also found in PCA and heatmap analysis in the present study. However, Σn-9 derived volatiles was positively correlated with the dietary lipid level instead of the contents of n-9 fatty acids in the present study. Could it be that more n-9 fatty acids in high dietary lipid level groups were degraded before those were measured? Further study was needed.

### 4.4. Fillet Taste

Taste is an important flavor characteristic for consumer satisfaction together with odor [6]. The taste profile was related to five basic flavors of umami, sweetness, bitterness, saltiness and sourness, which can be recognized by the non-volatile compounds [36]. Umami is the main taste characteristic for aquatic products, which can be evaluated by three nucleotides (IMP, AMP and GMP) and two free amino acids (Free asp and Free glu) concentrations [19]. Except for Free glu, fish fed the diet with a low lipid level (6.6%) had the lowest concentrations of umami-related taste compounds, while the SUTC increased as the dietary lipid level increased up to 22.8% in the present study. EUC was very useful to evaluate the umami taste of aquatic products [19,23]. In the present study, 22.8% and more dietary lipid levels could improve flesh umami taste of triploid rainbow trout on the basis of EUC. The reason was related to purine metabolism and alanine, aspartate and glutamate metabolism on the basis of the enrichment analysis in this study. Besides, Free asp and Free glu were associated with umami taste, Free thr, ser, gly, ala, pro and lys were associated with sweet taste and Free arg, val, met, leu, iso, his, phe and tyr were associated with bitter taste [19]. In the present study, the SWTC in the fillet of triploid rainbow trout was not affected by the dietary lipid level, while the 22.8% and more dietary lipid level could decrease the flesh bitter taste on the basis of SBTC.

Organic acids could contribute a special sour taste, such as succinic acid and lactic acid [19,21]. This study firstly identified the organic acids species in the fillet of triploid rainbow trout by the retention times of 21 species of the organic acid standards kit. Only lactic acid, succinic acid, oxalic acid and maleic acid were found. Based on the sum concentration of four organic acids, fish fed the diet with the lowest lipid level (6.6%) had the highest sour taste. Based on PCA and heatmap analysis, the fillet SOTC content positively correlated with fillet glycogen content. As mentioned above, fish fed the diet with a low lipid level had high muscle glycogen content, which might be related to a high level of glycogen degradation and lactate accumulation and then resulted in the fillet with the low pH and sour taste. Salts composed of cations (Na^+^, K^+^, Ca^2+^, Mg^2+^) and anions (Cl^−^, PO_4_^3−^) contributed to the salty taste, such as sodium chloride. In the present study, the concentration and the sum concentration of those inorganic ions were not affected by dietary lipid levels, which suggested that flesh saltiness of triploid rainbow trout was not affected. In total, the high lipid diet (≥22.8%) could improve the taste of triploid rainbow trout to increase umami taste as well as decrease the bitter and sour taste.

Similar to OAV, TAV was a useful index to evaluate the food taste, because it took both the concentration and taste threshold [19]. Based on the data of TAVs, taste active compounds (TAV ≥ 1) in the fillet of triploid rainbow trout were IMP, free glu, free his, lactic acid, succinic acid, oxalic acid, K^+^ and PO_4_^3−^, which needed attention in the following quality study of triploid rainbow trout. In those, IMP, lactic acid, succinic acid and oxalic acid contents were affected by the dietary lipid level.

### 4.5. Fillet Nutritional Value

Protein is a fundamental nutrient, fish and seafood products are one of the most important sources of dietary protein for humans [5]. In the present study, the protein content of triploid rainbow trout ranged from 205 to 215 g/kg muscle, which decreased with the increase of the dietary lipid level. A similar trend was also found in another study [8]. Except for protein content, the essential amino acid composition (calculated as amino acid score) is also of vital importance for nutritional qualities of protein [21]. All EAAS of fillet were > 100 for all EAAs in the present study, which meant that triploid rainbow trout represented a good protein source with high nutritional value. The ratio of EAA/TAA ranged between 0.49–0.50 in the present study, which was higher than the ideal ratio of EAA/TAA (approximately 0.4) recommended by the FAO/WHO/UNU (1985) [22]. It also meant that the fillets in all treatments had high nutritional value for amino acid composition and were more beneficial to human health. Additionally, Trp was the least amino acid in the fillet of triploid rainbow trout, and its content decreased with an increasing level of dietary lipid up to 22.8%. Based on PCA and heatmap analysis, Trp content negatively correlated with fillet lipid content. Trp is proven to be an important precursor of serotonin, melatonin and niacin [37]. The previous study showed that niacin was related to lipid metabolism in fish [38]. In the present study, Trp content decreased in high lipid levels groups (22.8% and 29.4%). The reason could be that Trp acted as a precursor participating in lipid metabolism. Hence, a higher Trp level should be supplemented in the diet with a high lipid level.

Lipids are necessary for energy supplying and maintaining the normal functions of the human body [39]. In the present study, fillet lipid content increased with an increased dietary lipid level. The positive correlation was also found in other studies [8,11]. Fatty acids are the main constituent of lipid and essential fatty acids and are more important for human health. Generally, linoleic acid (18:2n-6) and α-linolenic acid (18:3n-3) are two essential fatty acids for humans [26]. In the present study, linoleic acid and α-linolenic acid content in the fillet of triploid rainbow trout increased with an increased dietary lipid level up to 22.8% and 29.4%, respectively. In addition, it is known that EPA and DHA are benefits not only in growth and development of newborns but also in the prevention of several disease, such as chronic cardiovascular, diabetes, cancer and age-related degenerative diseases [40]. In the present study, EPA + DHA content also increased with an increased dietary lipid level. Based on the result of enrichment analysis, a high lipid level (≥22.8%) could improve α-linolenic acid and linoleic acid metabolism and biosynthesis of unsaturated fatty acids. In addition, a previous study showed that the fatty acids profile of the fillet could be affected by the dietary lipid source [35]. The increased α-linolenic acid and EPA + DHA contents of the fish fillet in the present study could attribute to using fish oil as the main dietary lipid source in the diets. AI and TI value were used to evaluate the nutritional quality of fat in food [16]. They measured the potential capacity of the fats, in particularly sensitive people, to provoke atheromae and thrombi, respectively, and concluded that very low values of AI and TI were recommended [39]. In the present study, the AI and TI value ranged between 0.27–0.31 and 0.36–0.42, respectively, which were far below those in lamb (AI: 1.0; TI: 1.33–1.58), beef (AI: 0.70–0.74; TI: 0.79–1.39) and pork (AI: 0.58–0.60; TI: 1.35–1.66) [16]. Besides, the AI value further decreased in the 22.8% and 29.4% lipid level groups under the present condition, which suggested that diets with high fish oil supplemented could improve the nutritional quality of triploid rainbow trout.

Minerals are important micronutrients for humans due to potential health enhancement. In the present study, nine minerals, including Na, K, Ca, Mg, Mn, Fe, Cu, Zn and Se were analyzed, in which only Se content was affected. In this study, fillet Se content was the lowest in the highest dietary lipid level group. The reason might be related to the high LC-PUFA content in fish fed the diet with the 29.4% lipid level, as LC-PUFAs were highly susceptible to oxidation. Se contains a strong antioxidant agent, such as selenoprotein, which protects the cellular component, including cell membrane against oxidative damage [41]. In the present study, some Se was used for antioxidation rather than deposited in the fillet. Hence, a higher Se level should be supplemented in the diet with a high lipid level.

Based on the data of nutritional contribution % DRI or DAI in the present study, the top five nutrients in the fillet of triploid rainbow trout that contribute most to human nutrition were EPA + DHA (NC: 89.7–101.9%), protein (NC: 68.6–71.9%), Se (NC: 32.6–53.2%), K (NC: 25.4–27.3%) and Zn (NC: 9.87–14%). The NC value of the nutrients was all more than 10%, which should be focused on further to evaluate the quality of triploid rainbow trout.

## 5. Conclusions

Based on the results of 139 physico-chemical indicators and multivariate analyses, the dietary lipid level could affect the organoleptic and nutritional quality of triploid rainbow trout in the present conditions. Generally, high lipid diets (dietary lipid level were 22.8 and 29.4%) could raise fillet redness, enhance the odor intensity of “green, fatty and fishy”, decrease the bitterness and sourness taste, increase the umami taste and lipid nutritional value. Hence, it was suggested that high lipid diets could not only promote growth but also improve the quality of triploid rainbow trout. The recommended minimum dietary lipid level for triploid rainbow trout should be 22.8% to keep good organoleptic and nutritional quality. Further studies are needed to specify how feed nutrition regulates the physico-chemical parameters related to fish quality.

## Figures and Tables

**Figure 1 foods-12-00015-f001:**
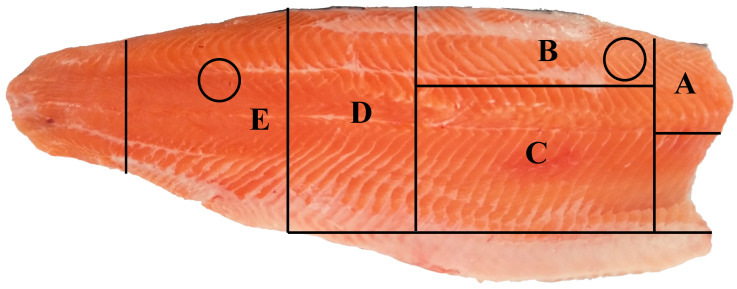
Sampling segments for measurements of different quality indices in fillet of triploid rainbow trout. Circular areas were used to assay fillet color, texture and pH; Segment A was used to assay water holding capacity; Segment B was used to assay water/salt soluble protein, hydroxyproline, pyridinoline crosslink and glycogen; Segment C was used to assay free amino acids, nucleotides, organic acids and inorganic ions; Segment D was used to assay lipid, fatty acids, amino acids and volatile compounds; Segment E was used to assay moisture, ash, total pron.

**Figure 2 foods-12-00015-f002:**
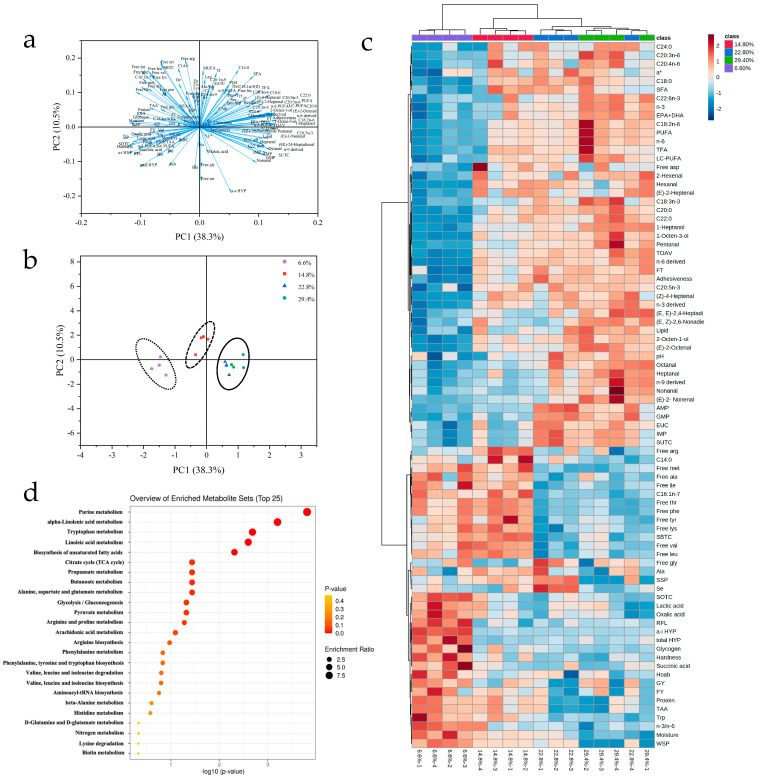
Multivariate statistical analysis and enrichment analysis of fillet quality indicators in different dietary lipid level groups. Principal component analysis (PCA) and cluster analysis of the results from fish samples showing the first two principal components (PC1 and PC2). In the correlation loading plot of the measured variables (**a**), there are 139 arrows and each arrow point to one quality variable (index); In the score plot (**b**), there are four pooled samples within each dietary lipid level group, each pooled sample consisted of three fish (*n* = 4; *N* = 12). Heatmap analysis of fillet quality indicators with significant difference in 6.6, 14.8, 22.8 and 29.4% dietary lipid level group (**c**); *n* = 4; *N* = 12). Enrichment analysis diagram of the different metabolic pathways between the group with dietary lipid level higher than or equal to 22.8% and the group with dietary lipid level lower than 22.8% (**d**).

**Table 1 foods-12-00015-t001:** Effects of dietary lipid levels on biometrical parameters and fillet color of triploid rainbow trout fed the experimental diets for 80 days.

	Dietary Lipid Levels (% Dry Matter)			
	6.6	14.8	22.8	29.4
Biometrical parameters				
Gutted yield (GY, %) ^1^	88.2 ± 0.3 ^a^	86.9 ± 0.4 ^ab^	86.5 ± 0.4 ^b^	86.4 ± 0.4 ^b^
Fillet yield (FY, %) ^1^	65.1 ± 0.3 ^a^	64.4 ± 0.3 ^ab^	63.9 ± 0.3 ^b^	63.7 ± 0.4 ^b^
Relative fillet length (RFL) ^1^	0.806 ± 0.006 ^a^	0.777 ± 0.004 ^b^	0.779 ± 0.003 ^b^	0.778 ± 0.003 ^b^
Fillet thickness (FT, mm) ^2^	10.9 ± 0.5 ^b^	12.6 ± 0.6 ^ab^	13.3 ± 0.5 ^a^	13.2 ± 0.5 ^a^
Fillet color				
*L** ^3^	45.2 ± 0.6	44.2 ± 0.4	44.8 ± 0.9	45.6 ± 0.5
*a** ^3^	14.2 ± 0.5 ^b^	15.3 ± 0.3 ^ab^	15.8 ± 0.6 ^a^	15.6 ± 0.4 ^a^
*b** ^3^	20.5 ± 0.7	20.6 ± 0.3	20.8 ± 0.6	20.6 ± 0.6
*C*_ab_* ^4^	25.1 ± 0.8	25.4 ± 0.4	25.9 ± 0.9	26.1 ± 0.6
$H_{\mathrm{ab}}^{0}$ ^4^	55.6 ± 0.6 ^a^	53.4 ± 0.3 ^ab^	52.7 ± 0.8 ^b^	53.0 ± 0.5 ^b^

Values are shown as mean ± standard error (*n* = 4, *N* = 12) and different superscript letters in the same row indicate significantly (*p* < 0.05) among dietary lipid level groups. ^1^ Gutted yield (GY, %) = 100 × (carcass gutted weight (g)/body weight (g)); Fillet yield (FY, %) = 100 × (fillets weight (g)/body weight (g)); Relative fillet length = fillet length (cm)/body length (cm). ^2^ Fillet thickness (FT) was assayed by texture analyzer on the basis of the texture profile analyses (TPA). ^3^
*L**, *a**, *b**, *C*_ab_* and $H_{\mathrm{ab}}^{0}$ represent lightness, redness, yellowness, respectively. ^4^
*C*_ab_* (chroma) = (*a**^2^ + *b**^2^)^1/2^; $H_{\mathrm{ab}}^{0}$ (hue) = arctan (*b**/*a**).

**Table 2 foods-12-00015-t002:** Effects of dietary lipid levels on fillet texture of triploid rainbow trout fed the experimental diets for 80 days.

	Dietary Lipid Levels (% Dry Matter)			
	6.6	14.8	22.8	29.4
Physical properties				
Texture profile analyses (TPA)				
Fracturability (N)	4.15 ± 0.17	4.43 ± 0.21	4.16 ± 0.14	3.94 ± 0.15
Hardness (N)	5.59 ± 0.21 ^a^	4.94 ± 0.15 ^b^	4.71 ± 0.13 ^b^	4.60 ± 0.10 ^b^
Adhesiveness (mJ)	1.98 ± 0.07 ^c^	2.72 ± 0.08 ^b^	2.90 ± 0.06 ^ab^	3.01 ± 0.06 ^a^
Cohesiveness	0.18 ± 0.01	0.18 ± 0.01	0.18 ± 0.01	0.17 ± 0.01
Springiness (mm)	3.03 ± 0.35	3.35 ± 0.32	3.44 ± 0.22	3.24 ± 0.15
Chewiness (mJ)	2.51 ± 0.22	2.83 ± 0.23	2.98 ± 0.28	2.53 ± 0.10
Water holding capacity (WHC, %)	92.3 ± 0.2	92.1 ± 0.4	92.4 ± 0.3	91.4 ± 0.4
pH	6.24 ± 0.02 ^b^	6.27 ± 0.01 ^ab^	6.31 ± 0.01 ^a^	6.30 ± 0.01 ^a^
Biochemical compositions				
Moisture (mg/g muscle)	728 ± 4 ^a^	715 ± 3 ^ab^	707 ± 5 ^bc^	696 ± 5 ^c^
Ash (mg/g muscle)	13.1 ± 0.4	12.4 ± 0.4	12.1 ± 0.4	11.7 ± 0.5
Lipid (mg/g muscle)	64.6 ± 4.9 ^c^	87.7 ± 6.1 ^b^	91.0 ± 5.5 ^ab^	105.7 ± 4.8 ^a^
Glycogen (mg/g muscle)	1.51 ± 0.22 ^a^	1.03 ± 0.13 ^ab^	0.80 ± 0.05 ^b^	0.73 ± 0.04 ^b^
Water soluble protein (WSP, mg/g muscle)	79.1 ± 4.1 ^a^	59.9 ± 1.0 ^b^	55.7 ± 3.1 ^b^	13.3 ± 0.2 ^c^
Salt soluble protein (SSP, mg/g muscle)	44.1 ± 7.7 ^b^	53.7 ± 2.0 ^ab^	65.5 ± 2.2 ^a^	25.2 ± 0.3 ^c^
Collagen				
a-s HYP (mg/g muscle) ^1^	0.12 ± 0.01	0.10 ± 0.01	0.13 ± 0.01	0.14 ± 0.01
a-i HYP (mg/g muscle) ^1^	0.23 ± 0.02 ^a^	0.01 ± 0.01 ^b^	0.01 ± 0.00 ^b^	0.01 ± 0.00 ^b^
Total HYP (mg/g muscle) ^1^	0.32 ± 0.03 ^a^	0.11 ± 0.01 ^b^	0.14 ± 0.01 ^b^	0.15 ± 0.01 ^b^
PYD (μg/g muscle) ^1^	2.08 ± 0.11	2.12 ± 0.08	2.00 ± 0.14	2.03 ± 0.09

Values are shown as mean ± standard error (*n* = 4, *N* = 12) and different superscript letters in the same row indicate significantly (*p* < 0.05) among dietary lipid level groups. ^1^ a-s HYP: alkaline-soluble hydroxyproline; a-i HYP: alkaline-insoluble hydroxyproline; total HYP: total hydroxyproline; PYD: pyridinoline crosslink.

**Table 3 foods-12-00015-t003:** Effects of dietary lipid levels on fillet flavor (odor and taste) of triploid rainbow trout fed the experimental diets for 80 days.

		Dietary Lipid Levels (% Dry Matter)			
	Flavor Description	6.6	14.8	22.8	29.4
Odor-active compounds ^1^	Odor description ^2^	ng/g muscle (OAV ^1^)			
1-Heptanol	Green, fermented, nutty	49.5 ± 1.7 (9.2) ^c^	72.7 ± 0.5 (13.5) ^b^	81.6 ± 4.6 (15.1) ^b^	94.8 ± 1.2 (17.6) ^a^
1-Octen-3-ol	Earthly, mushroom, fermented	254 ± 5 (169) ^c^	437 ± 17 (291) ^b^	473 ± 2 (315) ^ab^	528 ± 38 (352) ^a^
2-Octen-1-ol	Dirt, mushroom	30.1 ± 2.2 (0.75) ^b^	58.0 ± 3.1(1.45) ^a^	65.9 ± 2.2 (1.65) ^a^	67.6 ± 4.2 (1.69) ^a^
Pentanal	Acetaldehyde-like, pungent	78 ± 1 (8.7) ^c^	112 ± 5 (12.4) ^b^	123 ± 2 (13.7) ^b^	151 ± 12 (16.8) ^a^
Hexanal	Garlic, green, grassy, pungent, fatty, fishy	449 ± 7 (100) ^b^	913 ± 60 (203) ^a^	925 ± 26 (206) ^a^	796 ± 104 (177) ^a^
2-Hexenal	Moss, mushroom	11.1 ± 1.1 (0.58) ^b^	19.6 ± 2.0 (1.02) ^a^	19.5 ± 2.2 (1.01) ^a^	21.1 ± 1.9 (1.10) ^a^
Heptanal	Green, floral, fatty, pungent, fishy, nutty, mushroom	124 ± 18 (44.3) ^c^	166 ± 26 (59.4) ^bc^	224 ± 29 (80.0) ^ab^	291 ± 18 (103.8) ^a^
(*E*)-2-Heptenal	Roast meat, cooked fish, sulfury	15.1 ± 1.9 (1.16) ^b^	41.3 ± 5.5 (3.18) ^a^	39.7 ± 4.2 (3.05) ^a^	35.0 ± 5.3 (2.69) ^a^
(Z)-4-Heptenal	Fishy, boiled potato	59 ± 1 (13.9) ^b^	113 ± 9 (26.9) ^a^	120 ± 12 (28.6) ^a^	108 ± 7 (25.7) ^a^
(*E*, *E*)-2,4-Heptadienal	Fishy, grassy	14.2 ± 1.5 (0.92) ^c^	18.8 ± 1.1 (1.22) ^bc^	22.9 ± 2.7 (1.49) ^ab^	27.4 ± 1.9 (1.78) ^a^
Octanal	Sweet, orange, floral, pungent, green, fatty	85 ± 9 (122) ^b^	113 ± 5 (162) ^b^	161 ± 12 (230) ^a^	164 ± 9 (234) ^a^
(*E*)-2-Octenal	Moldy, pungent, cucumber, fatty, mushroom	18.4 ± 1.3 (6.1) ^c^	45.9 ± 2.5 (15.3) ^b^	47.5 ± 3.3 (15.8) ^b^	62.9 ± 4.2 (21.0) ^a^
Nonanal	Geranium, fishy, plastic, orange, green, fatty	175 ± 21 (159) ^b^	182 ± 14 (166) ^b^	266 ± 27 (242) ^ab^	343 ± 51 (312) ^a^
(*E*)-2- Nonenal	Moss, woody, floral, green, fruity	8.8 ± 0.5 (110) ^c^	16.3 ± 1.1(203) ^bc^	21.2 ± 2.9 (264) ^b^	36.4 ± 4.1 (454) ^a^
(*E, Z*)-2,6-Nonadienal	Cucumber, floral	29.5 ± 2.5 (36.8) ^b^	38.0 ± 2.9 (47.5) ^ab^	37.2 ± 2.6 (46.5) ^ab^	46.9 ± 0.3 (58.6) ^a^
Total (TOAV)		1400 ± 46 ^c^	2346 ± 82 ^b^	2628 ± 79 ^ab^	2773 ± 173 ^a^
∑n-3 derived ^3^		113 ± 5 ^b^	189 ± 5 ^a^	200 ± 17 ^a^	204 ± 7 ^a^
∑n-6 derived ^4^		977 ± 19 ^b^	1790 ± 71 ^ab^	1919 ± 33 ^a^	1968 ± 118^a^
∑n-9 derived ^5^		434 ± 47 ^c^	534 ± 30 ^bc^	732 ± 69 ^ab^	893 ± 72^a^
Taste-related compounds	Taste description ^6^	mg/100 g muscle (TAV ^7^)			
AMP ^8^	Umami	2.14 ± 0.24 (0.04) ^d^	3.64 ± 0.13 (0.08) ^c^	10.23 ± 0.32 (0.21) ^a^	6.45 ± 0.56 (0.13) ^b^
IMP ^8^	Umami	81.1 ± 16.2 (3.25) ^c^	126.2 ± 20.2 (5.05) ^bc^	225.5 ± 16.9 (9.02) ^a^	193.0 ± 22.7 (7.73) ^ab^
GMP ^8^	Umami	0.17 ± 0.02 (0.01) ^c^	0.26 ± 0.03 (0.02) ^c^	0.53 ± 0.02 (0.04) ^a^	0.41 ± 0.03 (0.03) ^b^
Free aspartic acid (Free asp)	Umami	0.21 ± 0.01 (0.002) ^b^	0.30 ± 0.03 (0.003) ^a^	0.26 ± 0.01 (0.003) ^ab^	0.25 ± 0.02 (0.003) ^ab^
Free glutamic acid (Free glu)	Umami	29.9 ± 1.1 (1.00)	29.4 ± 0.7 (0.98)	24.7 ± 0.6 (0.82)	27.2 ± 2.6 (0.91)
Sum of umami taste compounds (SUTC)		114 ± 17^c^	160 ± 21 ^bc^	261 ± 17 ^a^	227 ± 21 ^ab^
EUC (g/100g) ^9^		3.07 ± 0.68 ^b^	4.64 ± 0.84 ^ab^	6.90 ± 0.48 ^a^	6.31 ± 0.40^a^
Free threonine (Free thr)	Sweet	2.35 ± 0.16 (0.01) ^a^	2.53 ± 0.07 (0.01) ^a^	1.34 ± 0.06 (0.01) ^b^	1.51 ± 0.10 (0.01) ^b^
Free serine (Free ser)	Sweet	7.37 ± 0.41 (0.05)	6.71 ± 0.23 (0.04)	7.82 ± 0.58 (0.05)	7.00 ± 0.57 (0.05)
Free glycine (Free gly)	Sweet	69.3 ± 3 (0.53) ^ab^	62.4 ± 1.6 (0.50) ^b^	74.4 ± 2.5 (0.57) ^a^	65.3 ± 3.8 (0.50) ^ab^
Free alanine (Free ala)	Sweet	41.4 ± 0.9 (0.69) ^a^	41.0 ± 0.3 (0.69) ^ab^	38.5 ± 0.3 (0.64) ^b^	39.4 ± 0.7 (0.66) ^ab^
Free proline (Free pro)	Sweet	9.72 ± 1.68 (0.03)	9.09 ± 0.56 (0.03)	5.07 ± 1.66 (0.02)	7.22 ± 2.12 (0.02)
Free lysine (Free lys)	Sweet	1.67 ± 0.06 (0.03) ^a^	1.75 ± 0.03(0.04) ^a^	1.33 ± 0.05(0.03) ^b^	1.47 ± 0.03(0.03) ^b^
Sum of sweet taste compounds (SWTC)		132 ± 4	124 ± 2	129 ± 4	122 ± 5
Free argnine (Free arg)	Bitter	2.89 ± 0.08 (0.06) ^b^	3.75 ± 0.12 (0.08) ^a^	2.86 ± 0.20 (0.06) ^b^	2.74 ± 0.28 (0.06) ^b^
Free valine (Free val)	Bitter	13.4 ± 0.7 (0.34) ^ab^	14.4 ± 0.2 (0.36) ^a^	10.2 ± 0.4 (0.26) ^c^	11.5 ± 0.4 (0.29) ^bc^
Free methionine (Free met)	Bitter	6.00 ± 0.33 (0.20) ^a^	6.11 ± 0.31 (0.20) ^a^	4.42 ± 0.17 (0.15) ^b^	4.32 ± 0.20 (0.15) ^b^
Free leucine (Free leu)	Bitter	12.9 ± 0.7 (0.07) ^a^	13.7 ± 0.2 (0.07) ^a^	9.8 ± 0.4 (0.05) ^b^	10.7 ± 0.5 (0.06) ^b^
Free isoleucine (Free iso)	Bitter	4.82 ± 0.3 (0.06) ^a^	4.54 ± 0.08 (0.05) ^a^	3.60 ± 0.12 (0.04) ^b^	3.76 ± 0.12 (0.04) ^b^
Free histidine (Free his)	Bitter	61.0 ± 1.1 (3.05)	64.9 ± 0.9 (3.24)	62.4 ± 0.9 (3.12)	65.2 ± 1.5 (3.26)
Free phenylalanine (Free phe)	Bitter	8.02 ± 0.53 (0.09) ^a^	8.57 ± 0.26 (0.10) ^a^	4.63 ± 0.18 (0.05) ^b^	5.22 ± 0.35 (0.06) ^b^
Free tyrosine (Free tyr)	Bitter	9.99 ± 0.36 (-) ^ab^	10.67 ± 0.51 (-) ^a^	8.49 ± 0.21 (-) ^c^	9.01 ± 0.26 (-) ^bc^
Sum of bitter taste compounds (SBTC)		119 ± 2^ab^	127 ± 1 ^a^	106 ± 1 ^c^	113 ± 3 ^bc^
Lactic acid	Sour	649 ± 28 (5.15) ^a^	535 ± 33 (4.24) ^ab^	499 ± 35 (3.97) ^b^	521 ± 36 (4.14) ^ab^
Succinic acid	Sour	406 ± 39 (38.33) ^a^	271 ± 18 (25.52) ^b^	272 ± 29 (25.68) ^b^	235 ± 19 (22.11) ^b^
Oxalic acid	Sour	401 ± 17 (7.96) ^a^	330 ± 15 (6.54) ^b^	292 ± 20 (5.79) ^b^	306 ± 15 (6.06) ^b^
Maleic acid	Sour	11.8 ± 0.8 (-)	10.9 ± 0.9 (-)	11.4 ± 0.4 (-)	12.7 ± 0.3 (-)
Sum of sour taste compounds (SOTC)		1468 ± 39 ^a^	1146 ± 63 ^b^	1075 ± 62 ^b^	1074 ± 35 ^b^
Na^+^	Saline	35.9 ± 0.9 (0.20)	34.0 ± 2.9 (0.19)	41.6 ± 2.7 (0.23)	32.7 ± 2.4 (0.18)
K^+^	Saline	475 ± 8 (3.65)	482 ± 26 (3.71)	486 ± 7 (3.74)	451 ± 21 (3.47)
Ca^2+^	Saline	13.8 ± 1.2 (0.09)	18.0 ± 2.9 (0.12)	11.9 ± 0.9 (0.08)	17.3 ± 4.5 (0.12)
Mg^2+^	Saline	31.7 ± 0.5 (0.33)	32.7 ± 1.9 (0.34)	33.1 ± 1.5 (0.35)	30.0 ± 1.2 (0.31)
Cl^−^	Saline	47.6 ± 0.7 (0.37)	45.9 ± 1.4 (0.35)	47.6 ± 4.8 (0.37)	39.0 ± 3.1 (0.30)
PO_4_^3−^	Saline	512 ± 24 (3.94)	576 ± 13 (4.43)	520 ± 25 (4.00)	577 ± 12 (4.44)
Sum of salty taste compounds (SATC)		1116 ± 17	1189 ± 28	1140 ± 27	1147 ± 28

Values are shown as mean ± standard error (*n* = 4, *N* = 12) and different superscript letters in the same row indicate significantly (*p* < 0.05) among dietary lipid level groups. ^1^ Odor-active compounds are volatile compounds with the odor activity values (OAVs, the ratio of concentration and odor threshold of volatile compound) equal to or greater than 1 [17]. ^2^ Odor description from literature: Ma et al. [17]. ^3^ ∑n-3 derived: Sum of 2-hexenal, (Z)-4-heptenal, (*E*, *E*)-2,4-heptadienal, (*E*, *Z*)-2,6-nonadienal, which are derived by n-3 fatty acids. ^4^ ∑n-6 derived: Sum of 1-octen-3-ol, 2-octen-1-ol, pentanal, hexanal, heptanal, (*E*)-2-heptenal, (*E*)-2-octenal, (*E*)-2- nonenal, which are derived by n-6 fatty acids. ^5^ ∑n-9 derived: Sum of 1-heptanol, heptanal, octanal, nonanal, which are derived from n-9 fatty acids. ^6^ Taste description from literature: Liu et al. [23]. ^7^ TAV: the taste activity value, the ratio of concentration and taste threshold of the compound. ^8^ AMP: Adenosine-5′- monophosphate; GMP: guanosine-5′-monophosphate; IMP: inosine-5′-monophosphate. ^9^ EUC: the equivalent umami concentration.

**Table 4 foods-12-00015-t004:** Effects of dietary lipid levels on fillet nutrition value of triploid rainbow trout fed the experimental diets for 80 days.

	Dietary Lipid Levels (% Dry Matter)			
	6.6	14.8	22.8	29.4
Protein	g/kg muscle (Nutritional contribution% DRI ^1^)			
	215 ± 1 (71.9) ^a^	212 ± 2 (71.1) ^ab^	206 ± 2 (69.0) ^bc^	205 ± 2 (68.6) ^c^
Amino acids after hydrolysis	g/kg muscle (Essential amino acids score ^2^)			
Histidine	13.23 ± 2.43 (410)	9.61 ± 0.19 (302)	9.79 ± 0.31 (317)	9.58 ± 0.51 (312)
Isoleucine	9.72 ± 0.30 (151)	10.31 ± 0.04 (162)	9.68 ± 0.14 (157)	9.08 ± 0.46 (148)
Leucine	16.90 ± 0.72 (133)	18.09 ± 0.06 (145)	17.31 ± 0.17 (142)	16.90 ± 0.49 (140)
Lysine	19.24 ± 0.32 (199)	19.84 ± 0.05 (208)	19.58 ± 0.28 (211)	18.65 ± 0.73 (202)
Methionine + Cystine	5.89 ± 0.94 (125)	4.65 ± 1.56 (100)	5.13 ± 1.25 (113)	4.59 ± 1.02 (102)
Phenylalanine + Tyrosine	17.11 ± 0.75 (209)	18.45 ± 0.08 (229)	18.24 ± 0.31 (233)	18.73 ± 0.63 (240)
Threonine	9.63 ± 0.07 (195)	9.78 ± 0.05 (201)	9.60 ± 0.16 (203)	9.96 ± 0.07 (211)
Valine	12.31 ± 1.15 (147)	11.60 ± 0.16 (140)	11.43 ± 0.32 (142)	10.58 ± 0.34 (132)
Tryptophan	2.48 ± 0.20 (192) ^a^	2.13 ± 0.12 (167) ^ab^	1.60 ± 0.10 (129) ^b^	1.61 ± 0.03 (131) ^b^
EAA ^3^	106.49 ± 1.50	104.43 ± 1.25	102.36 ± 0.12	99.67 ± 0.77
Alanine	12.96 ± 0.41 ^a^	13.80 ± 0.12 ^a^	13.60 ± 0.38 ^b^	12.62 ± 0.36 ^b^
Argnine	14.33 ± 1.16	12.95 ± 0.23	12.13 ± 0.31	12.45 ± 0.18
Aspartic acid	22.12 ± 0.44	22.81 ± 0.19	22.17 ± 0.40	22.31 ± 0.55
Glutamic acid	30.84 ± 0.33	31.00 ± 0.16	30.31 ± 0.75	30.68 ± 0.39
Glycine	10.65 ± 0.23	10.63 ± 0.13	10.55 ± 0.35	10.26 ± 0.38
Serine	8.44 ± 0.05	8.52 ± 0.11	8.36 ± 0.16	8.82 ± 0.09
Proline	8.14 ± 0.50	7.44 ± 0.44	6.59 ± 0.49	7.54 ± 0.27
NEAA ^3^	107.48 ± 0.82	107.13 ± 0.95	103.69 ± 2.35	104.66 ± 1.63
TAA ^3^	213.97 ± 1.16 ^a^	211.56 ± 1.30 ^a^	206.05 ± 2.27 ^b^	204.33 ± 1.97 ^b^
EAA/TAA	0.50 ± 0.01	0.49 ± 0.00	0.50 ± 0.01	0.49 ± 0.01
Fatty acids	g/kg muscle (Nutritional contribution% DRI or DAI ^1^)			
C12:0	0.01 ± 0.00	0.01 ± 0.00	0.01 ± 0.00	0.01 ± 0.00
C14:0	0.47 ± 0.02 ^ab^	0.55 ± 0.04 ^a^	0.45 ± 0.02 ^ab^	0.42 ± 0.01 ^b^
C16:0	4.15 ± 0.16	5.19 ± 0.41	4.92 ± 0.24	4.82 ± 0.24
C18:0	1.62 ± 0.05 ^b^	2.36 ± 0.14 ^a^	2.52 ± 0.10 ^a^	2.48 ± 0.13 ^a^
C20:0	0.07 ± 0.00 ^c^	0.11 ± 0.01 ^b^	0.13 ± 0.00 ^ab^	0.14 ± 0.01 ^a^
C22:0	0.04 ± 0.00 ^b^	0.09 ± 0.01 ^a^	0.09 ± 0.00 ^a^	0.10 ± 0.01 ^a^
C24:0	0.01 ± 0.00 ^b^	0.02 ± 0.00 ^ab^	0.03 ± 0.00 ^a^	0.03 ± 0.00 ^a^
C16:1n-7	1.40 ± 0.09 ^a^	1.49 ± 0.12 ^a^	0.83 ± 0.04 ^b^	0.85 ± 0.06 ^b^
C18:1n-9 (*Z*)	7.86 ± 0.40	8.31 ± 0.35	8.32 ± 0.29	8.70 ± 0.53
C18:1n-9 (*E*)	1.15 ± 0.10	0.99 ± 0.05	0.93 ± 0.05	0.97 ± 0.06
C20:1n-9	0.58 ± 0.03	0.62 ± 0.05	0.60 ± 0.03	0.59 ± 0.03
C22:1n-9	0.26 ± 0.04	0.30 ± 0.03	0.37 ± 0.02	0.36 ± 0.03
C24:1n-9	0.07 ± 0.01	0.08 ± 0.01	0.07 ± 0.01	0.07 ± 0.00
C18:2n-6	4.61 ± 0.16 (5.63) ^b^	6.59 ± 0.55 (8.05) ^ab^	7.43 ± 0.30 (9.07) ^a^	8.63 ± 0.85 (10.54) ^a^
C20:2n-6	0.33 ± 0.05	0.45 ± 0.05	0.49 ± 0.04	0.51 ± 0.07
C20:3n-6	0.18 ± 0.02 ^b^	0.33 ± 0.06 ^ab^	0.35 ± 0.03 ^ab^	0.48 ± 0.07 ^a^
C20:4n-6	0.15 ± 0.01 ^b^	0.21 ± 0.02 ^ab^	0.20 ± 0.02 ^ab^	0.23 ± 0.01 ^a^
C18:3n-3	0.06 ± 0.01 (0.73) ^c^	0.10 ± 0.01 (1.26) ^bc^	0.13 ± 0.01 (1.58) ^b^	0.18 ± 0.02 (2.23) ^a^
C20:5n-3 (EPA)	0.48 ± 0.03 ^b^	0.59 ± 0.01 ^a^	0.59 ± 0.02 ^a^	0.59 ± 0.01 ^a^
C22:6n-3 (DHA)	2.33 ± 0.03 ^c^	2.41 ± 0.07 ^bc^	2.46 ± 0.02 ^b^	2.60 ± 0.01 ^a^
Total (TFA)	25.81 ± 0.41 ^b^	30.78 ± 1.74 ^ab^	30.89 ± 0.28 ^ab^	32.75 ± 1.91 ^a^
ΣSFA ^4^	6.37 ± 0.20 ^b^	8.32 ± 0.59 ^a^	8.14 ± 0.32 ^a^	8.00 ± 0.37 ^a^
ΣMUFA ^4^	11.31 ± 0.37	11.79 ± 0.50	11.11 ± 0.30	11.53 ± 0.58
ΣPUFA ^4^	8.13 ± 0.29 ^c^	10.67 ± 0.69 ^b^	11.65 ± 0.28 ^ab^	13.22 ± 0.96 ^a^
ΣLC-PUFA ^4^	3.47 ± 0.14 ^c^	3.98 ± 0.15 ^b^	4.09 ± 0.07 ^ab^	4.41 ± 0.15 ^a^
Σn-3 ^4^	2.86 ± 0.06 ^c^	3.10 ± 0.07 ^b^	3.18 ± 0.02 ^b^	3.36 ± 0.02 ^a^
Σn-6 ^4^	5.27 ± 0.23 ^c^	7.58 ± 0.65 ^b^	8.48 ± 0.28 ^ab^	9.85 ± 0.95 ^a^
Σn-9 ^4^	9.92 ± 0.45	10.30 ± 0.42	10.27 ± 0.28	10.69 ± 0.57
Σn-3/∑n-6	0.55 ± 0.01 ^a^	0.42 ± 0.04 ^b^	0.38 ± 0.01 ^b^	0.35 ± 0.03 ^b^
EPA + DHA	2.80 ± 0.06 (89.7) ^c^	3.00 ± 0.07 (95.9) ^b^	3.05 ± 0.02 (97.5) ^ab^	3.19 ± 0.01 (101.9) ^a^
AI ^5^	0.31 ± 0.02 ^a^	0.33 ± 0.01 ^a^	0.30 ± 0.02 ^b^	0.27 ± 0.01 ^b^
TI ^6^	0.36 ± 0.02	0.42 ± 0.02	0.40 ± 0.02	0.37 ± 0.00
Minerals	mg/kg muscle (Nutritional contribution % DRI)			
Na	359 ± 9 (3.96)	340 ± 29 (3.76)	416 ± 27 (4.59)	327 ± 24 (3.60)
K	4751 ± 83 (26.8)	4822 ± 264 (27.2)	4855 ± 71 (27.3)	4505 ± 207 (25.4)
Ca	138 ± 12 (1.92)	180 ± 29 (2.51)	303 ± 184 (4.22)	173 ± 45 (2.41)
Mg	317 ± 5 (14.6)	327 ± 19 (15.0)	331 ± 15 (15.2)	300 ± 12 (13.8)
Mn	0.19 ± 0.09 (1.49)	0.12 ± 0.02 (0.96)	0.27 ± 0.14 (2.18)	0.10 ± 0.01 (0.83)
Fe	2.75 ± 0.92 (4.20)	4.00 ± 1.67 (6.10)	3.65 ± 0.75 (5.57)	2.46 ± 0.94 (3.74)
Cu	0.37 ± 0.05 (6.84)	0.46 ± 0.08 (8.58)	0.40 ± 0.03 (7.40)	0.35 ± 0.03 (6.44)
Zn	5.79 ± 0.44 (9.95)	8.14 ± 1.31 (14.00)	6.18 ± 0.57 (10.63)	5.74 ± 0.20 (9.87)
Se	0.12 ± 0.01 (37.8) ^ab^	0.13 ± 0.02 (38.3) ^ab^	0.17 ± 0.02 (53.2) ^a^	0.11 ± 0.01 (32.6) ^b^

Values are shown as mean ± standard error (*n* = 4, *N* = 12) and different superscript letters in the same row indicate significantly (*p* < 0.05) among dietary lipid level groups. ^1^ DRI: daily recommended intake; DAI: daily adequate intake. ^2^ Essential amino acids score = 100 × one essential amino acid content in sample/one essential amino acid content in reference protein for adult maintenance [22]. ^3^ EAA: essential amino acids; NEAA: no-essential amino acids; TAA: total amino acids. ^4^ SFA: saturated fatty acids; MUFA: monounsaturated fatty acids; PUFA: polyunsaturated fatty acids; LC-PUFA: long chain polyunsaturated fatty acids; n-3: n-3 polyunsaturated fatty acids; n-6: n-6 polyunsaturated fatty acids; n-9: n-9 monounsaturated fatty acids. ^5^ AI: atherogenic index = (C12:0 + 4 × C14:0 + C16:0)/(ΣMUFA + ΣPUFA). ^6^ TI: thrombogenic index = (C14:0 + C16:0 + C18:0)/(0.5 × ΣMUFA + 0.5 × Σn-6 + 3 × Σn-3 + Σn-3/Σn-6).

## Data Availability

The data presented in this study are available on request from the corresponding author.

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
