# Peer review of "Does Dietary Lipid Level Affect the Quality of Triploid Rainbow Trout and How Should It Be Assessed?"

_foods, 2022, doi:10.3390/foods12010015_

Round 1

Reviewer 1 Report

Manuscript ID: foods-2041964 ASSESSMENT OF ORGANOLEPTIC AND NUTRITIONAL QUALITY FOR FILLET OF TRIPLOID RAINBOW TROUT AND EFFECTS OF DIETARY LIPID LEVELS

1. The title of this manuscript is "Assessment of organoleptic and nutritional quality for fillet of triploid rainbow trout and effects of dietary lipid levels". It is important that the authors indicate what they mean by organoleptic quality assessment

Usually, organoleptic determination is based on the smell and taste organs, the tongue and olfactory system. But, in this work the smell was measured as volatile compounds and they were assayed by gas chromatography-mass spectrometry. Relatively to taste, defined in this manuscript as Taste activity value (TAV), the values of free amino acids, 5′-nucleotides, organic acids and inorganic ions were used.

Thus, this aspect deserves a clarification of the terms used. Maybe instead of organoleptic the authors could consider the use of chemical quality and the title could be “Assessment of chemical and nutritional quality of triploid rainbow trout fillets and effects of dietary lipid levels”

2. It is indicated in the Introduction (line 65) that: …fish organoleptic properties (including appearance quality, texture, odor and taste) but in the section material and methods there is not any indication about the determination of appearance quality. Could the authors precise how this parameter was determined?

3. Words odor and color should be changed to odour and colour throughout the manuscript

4. In the Introduction (Line 63) it is mentioned that “Thus, the present study established a system with multiple indices …” What system are the authors considering. Please define what you mean.

Note, that in Conclusion (line 578) it is mentioned that “A digital system was established to evaluate the organoleptic and nutritional quality of triploid rainbow trout by analysing 139 indexes for physical properties and chemical compositions in the present study.” How was this digital system conceived and what correlations were established among indices?

5. It is more adequate to move Table 1 to the results section. Moreover, in the Results section it is indicated that the biometrical parameters and colour of fillet are shown in Table 2. But they are in Table 1. The numbering of tables should be reviewed

6. The title of 2.4 is “Texture and proximate composition analysis”, but lipid content is not there and other analysis than proximate composition (such as water and salt soluble protein (WSP and SSP) as well as collagen and glycogen) are included. Please change the title accordingly.

7. Relatively to texture (Line 292) it is mentioned fracture, cohesiveness, springiness, chewiness. Please consider to use Fracturability instead of Fracture.

8. How was estimated the concentration of each volatile compound in samples

9. In Discussion section (line 428) it appears for the first time “condition factor” how it was defined?

10. In line 456 it is mentioned that “pH was closely related to the post-mortem evolution of the flesh and played an important role for the texture of fish fillet”. But the authors did not follow the post-mortem evolution! Please explain.

11. Line 459 it is mentioned that “Thus, the result for higher flesh pH in 22.8% and 29.4% lipid level groups was related with flesh glycogen content.” There is any study supporting this statement? Moreover, such relation is only strong for the trout fed with 22.8 and 29.4%lipid.

12. Line 466, appears the sentence “Fresh fish odor originated from volatile aromas compounds [32]”. It seems out of context. Please explain.

13. Line 489, “Umami was the main taste characteristic for aquatic products, which can be evaluated by three nucleotides (IMP, AMP and GMP) and two free amino acids (Free asp and Free glu) concentrations [19].” Consider to use is instead of was.

14. Line 504, “This study firstly identified the organic acids species in the fillet of triploid rainbow trout from 21 organic acids”. Please rephrase the sentence.

15. In line 582, “important nutrients (ESSA,”. Which means ESSA?

16. GENERAL COMMENTS

It is a manuscript with many analytical results, but its discussion is not always clear. Thus, it is suggested to evaluate the results carefully and see which ones better correlate.

On the other hand, there are results that the authors link, but whose relationship seems to have little consistency.

The interpretation of the results shown in figure 2c is not easy. Thus, a further explanation of the results of the quality indicators with significant difference and the overall trend shown in Figure. 2c by heatmap analysis, could be more deeply discussed

Author Response

Dear reviewer,   We have checked the corresponde between the lines indicated in  reply text and the lines revised of the manuscript, and updated the cover letter which was consist of the responses to reviewers and the revised manuscript.
  1. Question: The title of this manuscript is "Assessment of organoleptic and nutritional quality for fillet of triploid rainbow trout and effects of dietary lipid levels". It is important that the authors indicate what they mean by organoleptic quality assessment. Usually, organoleptic determination is based on the smell and taste organs, the tongue and olfactory system. But, in this work the smell was measured as volatile compounds and they were assayed by gas chromatography-mass spectrometry. Relatively to taste, defined in this manuscript as Taste activity value (TAV), the values of free amino acids, 5′-nucleotides, organic acids and inorganic ions were used. Thus, this aspect deserves a clarification of the terms used. Maybe instead of organoleptic the authors could consider the use of chemical quality and the title could be “Assessment of chemical and nutritional quality of triploid rainbow trout fillets and effects of dietary lipid levels”

Answer: Thank you for your good advice. The study used not only chemical compositions but also physical properties (such as texture, color) to evaluate the quality of triploid rainbow trout. It is seemly not suitable that change “organoleptic quality” to “chemical quality”. Based on the advice of Reviewer 3, we rephrased the title as “Does dietary lipid level affect the quality of triploid rainbow trout and how should it be assessed?”. If you have a better suggestion, please let us know.

  1. Question: It is indicated in the Introduction (line 65) that: …fish organoleptic properties (including appearance quality, texture, odor and taste) but in the section material and methods there is not any indication about the determination of appearance quality. Could the authors precise how this parameter was determined?

Answer: Thank you for your question. The appearance quality in the study is consist of fillet biometrical parameters (including fillet yield, relative fillet length and fillet thickness) and fillet colour. Following the reviewer’s suggestion, we have revised it in the section of Materials and Methods of the revised manuscript (Lines 103-115).

  1. Question: Words odor and color should be changed to odour and colour throughout the manuscript

Answer: Thanks for your question. Following the reviewer’s suggestion, we have revised those in the revised manuscript (Lines 19, 31, 45, 60, 68, 93, 98, 111, 158, 160, 163, 268, 271, 302, 303, 312, 341, 343, 344, 433, 471, 471-492, 495, 595).

  1. Question: In the Introduction (Line 63) it is mentioned that “Thus, the present study established a system with multiple indices …” What system are the authors considering. Please define what you mean. Note, that in Conclusion (line 578) it is mentioned that “A digital system was established to evaluate the organoleptic and nutritional quality of triploid rainbow trout by analysing 139 indexes for physical properties and chemical compositions in the present study.” How was this digital system conceived and what correlations were established among indices?

Answer: Thank you for your good question. The expression of“system” and “digital system” is not accurate. We have revised it in the revised manuscript (Lines 66-69; Lines 592-594).

  1. Question: It is more adequate to move Table 1 to the results section. Moreover, in the Results section it is indicated that the biometrical parameters and colour of fillet are shown in Table 2. But they are in Table 1. The numbering of tables should be reviewed.

Answer: Thank you for your good question. We have revised those in the revised manuscript (Lines 268, 275, 284, 297, 304, 316, 341, 355, 395).

  1. Question: The title of 2.4 is “Texture and proximate composition analysis”, but lipid content is not there and other analysis than proximate composition (such as water and salt soluble protein (WSP and SSP) as well as collagen and glycogen) are included. Please change the title accordingly.

Answer: Thank you for your good question. We have revised it in the revised manuscript (Lines 116-117).

  1. Question: Relatively to texture (Line 292) it is mentioned fracture, cohesiveness, springiness, chewiness. Please consider to use Fracturability instead of Fracture.

Answer: Thank you for your good question. We have revised it in the revised manuscript (Line 285; Table 2).

  1. Question: How was estimated the concentration of each volatile compound in samples

Answer: Thank you for your question. The concentrations were quantified by the  ratio of the peak area to the internal standard (2, 4, 6-trimethyl-pyridine, Sigma, China) on the basis of our previous study (Ma et al., 2020). Briefly, The 3 g mixed muscle of each fish and 4.5 ml saturated NaCl solution were added into a 15 ml headspace vial. Then, the mixture was homogenized by an electric homogenizer (XHF-D, Xinzhi, China) on ice. After 15 μl 2, 4, 6-trimethyl-pyridine (91.7 ng/μl, Sigma, China) was added as an internal standard (I. S.). Volatile compounds of triploid rainbow trout were detected by the gas chromatography-mass spectrometry (GC–MS, QP2020, Shimadzu, Japan) analysis coupled with automated solid-phase microextraction (SPME) system (AOC-6000, CTC, Switzerland). The estimated concentration of each volatile compound in the sample was calculated as follows:

Ma, R., Liu, X., Tian, H., Han, B., Li, Y., Tang, C., Zhu, K., Li, C., Meng,     Y., 2020.  Odor-active volatile compounds profile of triploid rainbow trout with different marketable sizes. Aquaculture Reports. 17, 100312.

  1. Question: In Discussion section (line 428) it appears for the first time “condition factor” how it was defined?

Answer: Thank you for your question. Condition factor is an important indicator for fish morphology, which is calculated as:

        Condition factor (CF) = 100 × [body weight (g)] / [body length (cm)]3

        Fish with high CF value shows stocky and short body type. We have added some detail in the revised manuscript (Lines 430-431).

  1. Question: In line 456 it is mentioned that “pH was closely related to the post-mortem evolution of the flesh and played an important role for the texture of fish fillet”. But the authors did not follow the post-mortem evolution! Please explain.

Answer: Thank you for your question. Previous study has shown that pH was changed with the post-mortem degradation. In the study, the fillets in all treatments were stored in ice box for 48 h. During the time, we delivered fillets from breeding farm to laboratory. After 48 h, we measured pH values of fillets in a very short time. Hence, we think that the fillets in all treatment were under the same phase of the post-mortem degradation (48 h). In other words, the variation of pH value among different treatments was not affected by post-mortem evolution.

  1. Question: Line 459 it is mentioned that “Thus, the result for higher flesh pH in 22.8% and 29.4% lipid level groups was related with flesh glycogen content.” There is any study supporting this statement? Moreover, such relation is only strong for the trout fed with 22.8 and 29.4%lipid.

Answer: Thank you for your question. Previous study shown that reduction in pH was accompanied by enhanced glycogen degradation and lactate accumulation (Matarneh et al., 2017). In the study, higher glycogen and lactic acid contents were found in 6.6% lipid level group. It indicated that fish fed the diet with 6.6% lipid level had higher flesh glycogen content and resulted in higher lactic acid content by anaerobic glycolysis. It was also supported by the result of enrichment analysis that glycolysis and pyruvate metabolism were different between < 22.8% dietary lipid levels group and ≥22.8% dietary lipid levels group. Based on above, we have rephrased the discussion in the revised manuscript (Lines 463-470).

        Matarneh, S.K.; England, E.M.; Scheffler, T.L.; Yen, C.N.; Wicks, J.C.; Shi, H.; Gerrard, D.E. A mitochondrial protein increases glycolytic flux. Meat Science 2017, 133, 119-125, doi:10.1016/j.meatsci.2017.06.007.

  1. Question: Line 466, appears the sentence “Fresh fish odor originated from volatile aromas compounds [32]”. It seems out of context. Please explain.

Answer: Thank you for your question. The sentence is intended to indicate the relationship between fish odour and fillet volatile compounds. We have changed the sentence to “Volatile compounds determines the aroma/odour of food products [33]”, and updated the citation in the revised manuscript (Line 473).

Wei, H.; Wei, Y.; Qiu, X.; Yang, S.; Chen, F.; Ni, H.; Li, Q. Comparison of potent odorants in raw and cooked mildly salted large yellow croaker using odor-active value calculation and omission test: Understanding the role of cooking method. Food Chem. 2023, 402, 134015, doi:https://doi.org/10.1016/j.foodchem.2022.134015.

  1. Question: Line 489, “Umami was the main taste characteristic for aquatic products, which can be evaluated by three nucleotides (IMP, AMP and GMP) and two free amino acids (Free asp and Free glu) concentrations [19].” Consider to use is instead of was.

Answer: Thank you for your good question. We have revised it in the revised manuscript (Line 497).

  1. Question: Line 504, “This study firstly identified the organic acids species in the fillet of triploid rainbow trout from 21 organic acids”. Please rephrase the sentence.

Answer: Thank you for your good question. We have revised it in the revised manuscript (Line 512-513).

  1. Question: In line 582, “important nutrients (ESSA,”. Which means ESSA?

Answer: Thank you for your good question. It is EAAS not ESSA. Based on the advice of Reviewer 2, we have deleted it in the revised manuscript.

  1. Question: GENERAL COMMENTS: It is a manuscript with many analytical results, but its discussion is not always clear. Thus, it is suggested to evaluate the results carefully and see which ones better correlate. On the other hand, there are results that the authors link, but whose relationship seems to have little consistency. The interpretation of the results shown in figure 2c is not easy. Thus, a further explanation of the results of the quality indicators with significant difference and the overall trend shown in Figure. 2c by heatmap analysis, could be more deeply discussed.

Answer: Thank you for your good advice. Following reviewer’s suggestion, we have rephrased the discussion in the revised manuscript (Lines 426-428, 449-453, 454-456, 465-470, 486-488, 516-519, 544-551).

Reviewer 2 Report

In the present study “Organoleptic and nutritional quality evaluation for fillet of triploid rainbow trout and effects of dietary lipid levels" (id. foods-2041964) authors provide contribute to investigate organoleptic and nutritional quality evaluation indexes compositions of triploid rainbow trout were established. Besides, effects of dietary lipid levels (6.6%, 14.8%, 22.8% and 29.4%) on the quality of triploid rainbow trout were analyzed in the study. The manuscript must be improved there are poor language, and it is not very easy to follow. English language needs to be improved before the paper can be published. 

General comments: This paper addresses an important aspect and is of high for fish chain concern. However, I think the authors failed to provide all the necessary data and some parts need to be improved and better explained. First, the manuscript's material and method section, is not clear, it is necessary rewrite and improve (ex. sampling, lab procedures, etc..). The conclusion section should be rewrite. I suggest improving study with new conclusions and clearner M&M secion. My strong feeling is that the paper must have improved before pubblication. 

Author Response

Thanks for your advices.  Because we received your comments at the deadline, we have carefully revised our manuscript  based on the other Reviewers, including language,  title, M&M, results, discussion and conclusion.  More details are shown in attachment (response to reviewers) and our revised manuscript. If you have any questions and comments, please let us know. 

Reviewer 3 Report

The manuscript entitled: "Organoleptic and nutritional quality evaluation for fillet of triploid rainbow trout and effects of dietary lipid levels" is about systematic organoleptic and nutritional quality evaluation indexes consisting of 139 indexes for physical properties and chemical compositions of triploid rainbow trout. Also, the effects of dietary lipid levels on the quality of triploid rainbow trout were analyzed and reported. In general, the research is interesting, well-designed and fits the journal's aims and scopes. Some comments are needed to address before the final decision by the Editor as follows:

1- Title: Improve the title, make it more informative, and bring the "Why" and "How" of the research in the title.

2- Abstract: No need to number the results, try to incorporate some quantitative data into the results to make it more interesting for the readers.

3- Keywords: Choose keywords other than the main words in the title. It will improve the visibility of the article.

4- Introduction: Improve the literature review using recent publications.

5- Materials and Methods:

It is not common to mix the methodology with the results. You reported some results in the methodology (Table 1). Consider transferring to the results part.

Some methods are too detailed, you can make it short and informative.

6- Results and discussion:

A clear and well-established part of the manuscript.

7- Conclusion: It is too long. Justify the research hypothesis and recommend future research (if any).

8- References: Add some references from 2022-2023.

Author Response

  1. Question: Title: Improve the title, make it more informative, and bring the "Why" and "How" of the research in the title.

Answer: Thank you for your question. Based on the advices of two Reviewers, we    rephrased the title as “Does dietary lipid level affect the quality of triploid rainbow trout and how should it be assessed?”. If you have a better suggestion, please let us know.

  1. Question: Abstract: No need to number the results, try to incorporate some quantitative data into the results to make it more interesting for the readers.

Answer: Thanks for your question. We have rephrase the Abstract in the revised  manuscript (Lines 14-28).

  1. Question: Keywords: Choose keywords other than the main words in the title. It will improve the visibility of the article.

Answer: Thanks for your good question. We have revised those in the revised manuscript (Line 31).

  1. Question: Introduction: Improve the literature review using recent publications.

Answer: Thanks for your good question. We have improved the literatures in the    revised manuscript ([1], [5], [11]).

  1. Question: Materials and Methods: It is not common to mix the methodology with the results. You reported some results in the methodology (Table 1). Consider transferring to the results part. Some methods are too detailed, you can make it short and informative.

Answer: Thanks for your good question. We have transferred Table 1 to the results     part (Line 275), and rephrasedsome methods (Lines 166-168; Lines 212-220) in the revised manuscript.

  1. Question: Results and discussion: A clear and well-established part of the manuscript.

Answer: Thank you.

  1. Question: Conclusion: It is too long. Justify the research hypothesis and recommend future research (if any).

Answer: Thanks for your question. We have rephrased the Conclusion in the revised manuscript (Lines 592-601).

  1. Question: References: Add some references from 2022-2023.

Answer: Thanks for your question. We have improved references from 2022-2023 in the revised manuscript ([1], [5], [11], [33], [37], [41]).

Round 2

Reviewer 2 Report

Paper has been sufficiently corrected and modified

Author Response

Thanks for your encouragement